# The Validity of Prayer Importance Scale (PIS)

**Małgorzata Tatala** [1,*] and **Marcin Wojtasiński** [2]

1   Department of Developmental Psychology, The John Paul II (The Second) Catholic University of Lublin,
    20-950 Lublin, Poland
2   Department of Experimental Psychology, The John Paul II (The Second) Catholic University of Lublin,
    20-950 Lublin, Poland; marcin.wojtasinski@kul.pl
*   Correspondence: malgorzata.tatala@kul.pl

**Abstract:** Prayer is a central element of religiosity but research has focused primarily on distinguishing its types and analyzing its functional aspect. A particularly important issue is the subjective evaluation of prayer importance, which so far has not been reflected in the form of an independent psychometric tool. This is why the goal of the presented study was to develop Prayer Importance Scale (PIS) based on Tatala's definition of the concept. Two studies were conducted to verify reliability and validity of the tool. The proposed model was found to fit the data well. Correlations of PIS with basic parameters of religiosity: religious awareness, religious feelings, religious decisions, bond with a fellowship of believers, religious practices, religious morality, religious experience and forms of profession of faith were found to be significant. PIS can be a quick method providing information on the degree of religiosity and be used in participant selection in research studies.

**Keywords:** prayer; prayer importance; EFA; CFA





## 1. Introduction

Prayer occupies a central place in the life of a Christian, it is a dialogue between a person and God (Tatala 2009). Statistics show that about 94% of Poles declare that they belong to a religious denomination, 81% consider themselves believers, and 70% pray at least once a week (GUS 2018). There is a number of operationalizations of the term prayer, but most researchers agree that it represents spiritual communication with a power viewed as divine (Baesler 2003; James 1902). Many types of prayer are distinguished on the basis of its content, the most common are adoration, confession, thanksgiving, supplication, reception, and obligatory prayer (Whittington and Scher 2010; Kulpaczyński and Tatala 2000; Tatala 2000). Prayer can be internal, take a form of meditation, or external, be accompanied by singing, gestures, fixed formulas or a spontaneous activity. Being a rich and diversified field of a person's inner experience, prayer accompanies their thoughts and reveals their relation to people, things and the outside world (Tatala 2009). Ladd and Spilka (Ladd and Spilka 2002) indicate three directions of the influence of prayer resulting from cognitive connections made by the person who is praying: inwards (prayer helps with self-discovery), outwards (prayer improves the quality of social relations), as well as 'upwards' (focus on the relationship person-God).

Research on religiosity often involves comparisons of non-religious and religious people (Arli and Pekerti 2017; Chui and Cheng 2015; Harris et al. 2009). Sometimes, however, research is directed just at religious groups, especially when the topics are related to issues of faith and learning about how religious persons function, i.e., their motivation, cognitive, emotional and social aspects of their lives (Kostrubiec-Wojtachnio and Tatala 2014; Tatala and Walesa 2020). In these cases, religiosity is viewed as a multidimensional construct and understood as a personal and positive relationship of a person with God, which is realized through religious awareness, religious feelings, religious decisions, bond with a fellowship of believers, religious practices, religious morality, religious experience and forms of profession of faith (Walesa 2005).

Many studies use simple indicators of prayer such as frequency (How often do you pray? How many times a week do you pray?) or duration (On average, how long does it take you to pray?). Obtaining this basic and formal information is important, but it should be supplemented with precise qualitative and quantitative specifications. For example: what is the definition of prayer for the respondent, what type of prayer does the respondent practice most often, what does prayer entail for the respondent, and what place does prayer occupy in the hierarchy of importance are just a few questions that should be controlled for (Tatala and Walesa 2020). Levin (Levin 1996) proposes that prayer should not be viewed as a tool to fulfill one's religiosity, but rather understood as a lifestyle. Current research on prayer evolves primarily around developmental, motivational, and cognitive topics (Finney and Malony 1985; Krok 2009; Zarzycka et al. 2020).

One of present issues with prayer is its potential impact in healing (Benson et al. 2006; Hood et al. 2018; Sloan and Ramakrishnan 2006), as well as treating it as a coping strategy (Bänziger et al. 2008; Krause 2004). It has been observed that prayer can have both positive and negative effects on human functioning (Ladd and McIntosh 2008). For example, in a study by Lambert and colleagues (Lambert et al. 2010) it was found that people who pray were significantly more likely to forgive their partner, compared to non-prayers. McCullough and Larson (McCullough and Larson 1999) noted that prayer is often used as a means of coping with problems, especially when they are difficult and persistent. In contrast, in another study which examined different forms of prayer, it was observed that certain types of prayer (such as confession, supplication, or obligatory prayer) were practiced by individuals with low life quality and low life satisfaction. Other types of prayer (adoration, thanksgiving, and acceptance) were more often associated with high quality of life (Whittington and Scher 2010). This was supported by findings that link prayer to decreased negative emotions and increased empathy (Butler et al. 2002). Finally, some research associated complete and literal reliance in prayer (and e.g., ignoring one's medical condition) with negative consequences (Beach et al. 2008).

Although there is a growing interest in empirically testing effects of prayer on health and its effectiveness as a coping strategy, the topic is still under-explored (Breslin and Lewis 2008). In their review of the literature Dein and Littlewood (Dein and Littlewood 2008) note that the topic of prayer in the context of human religiosity is largely marginalized, although the importance of prayer in human life is indisputable.

There is a number of methods which can be used to study different aspects of prayer (Ladd and McIntosh 2008). Prayer is also sometimes included in questionnaires on human religiosity and considered as one of its dimensions (Huber and Huber 2012; Rydz et al. 2017). In Poland there has been only one attempt at validation of Huber's model of prayer (Bartczuk and Zarzycka 2020) which resulted in distinguishing six subscales: Gratitude/Adoration, Fear, Repine, Self-Directive, Cooperative, and Passive.

Being in the core of religiosity, prayer can be an excellent criterion for selecting subjects for more in-depth studies. By integrating formal criteria, such as frequency and duration of prayer, with its other parameters like awareness, feelings, decisions, bond, practices, morality, experience (Walesa 2005) and other aspects like quality of life, self-esteem, and personality traits a complete profile of a person's religiosity can be drawn (Walesa and Tatala 2020). The above premises prompted the authors of this study to create a method allowing for a quick diagnosis of the level of religiosity based on the obtained score (Tatala 2008). When selecting the items, it was important that they be neutral towards any religion. It is proposed that prayer importance is manifested in persisting in prayer regardless of external circumstances (e.g., lack of results, unhappiness, the feeling of distance from God), as well as treating it as one of coping strategies (Tatala 2009; Tatala and Walesa 2020). Therefore, the scale contains items reflecting these two concepts.

## 2. Method

To begin with, the Prayer Importance Scale (PIS) was created by four specialists with a theological and psychological background who were presented the definition of prayer

importance (including the authors of the article and two independent specialists—Czesław Walesa, a recognized and well-known Polish scientist in the field of psychology of religion, and Agata Grzywaczewska, a psychologist and expert on issues related to prayer). They formulated and evaluated 11 test items reflecting prayer importance. Six expert judges rated how adequately the statements described the prayer importance on a scale of 1 to 5. The reliability analysis performed by expert judges (psychology students with psychometrics background and familiarity with issues related to the psychology of religion) proved the items to be statistically significant (Kendall's W = 0.49, $\chi^2$ = 29.29, df = 10, $p < 0.01$). After calculating the mean score for each item (Table 1), the items with low fit indices were taken out of the measure. Only those test items with mean greater than or equal to 4 were included.

**Table 1.** Expert judges' rating of test items: Means, Medians, and Standard Deviations.

| Items | M | Me | SD |
|---|---|---|---|
| I persist in prayer even when I don't see the results. | 5.00 | 5.00 | 0.00 |
| Although other events force priority, I find time to pray. | 4.50 | 5.00 | 0.84 |
| Prayer helps me to stay with God in times of doubt about the purpose of life. | 3.67 | 4.00 | 0.52 |
| I trust God even though things don't always go my way. | 3.00 | 3.00 | 0.63 |
| Prayer helps me make sense of the suffering and hardships I experience. | 3.83 | 4.50 | 10.60 |
| Prayer brings me peace and hope in difficult moments of my life. | 3.33 | 3.00 | 0.52 |
| Even though I experience misfortune, I persist in prayer. | 4.67 | 5.00 | 0.52 |
| I persist in prayer, even when God seems distant. | 4.33 | 4.00 | 0.52 |
| I persist in prayer, even when I run out of words. | 3.83 | 4.00 | 0.75 |
| Even though I am aware of the bad things I have done, I do not give up on praying. | 4.33 | 4.00 | 0.52 |
| Daily prayer gives me strength to overcome difficulties. | 4.33 | 4.50 | 0.82 |

The final list of test items included:

1. I persist in prayer even when I don't see the results.
2. Although other events force priority, I find time to pray.
3. Even though I experience misfortune, I persist in prayer.
4. I persist in prayer, even when God seems distant.
5. Even though I am aware of the bad things I have done, I do not give up on praying.
6. Daily prayer gives me strength to overcome difficulties.

Reliability analysis of the tool was repeated with a new sample of expert judges and it proved to be statistically significant (Kendall's W = 0.62, $\chi^2$ = 18.57, df = 5, $p < 0.01$). The judges agreed on the adequacy of the test items in relation to the theoretical construct.

Subsequently, two studies were conducted: exploratory factor analysis and exploratory structural equation modelling in study 1, and confirmatory factor analysis carried out in study 2.

### 2.1. Study 1

The first study verified if the proposed model maintained its psychometric properties on the sample of 240 adults ($n_{women}$ = 113). Because four individuals did not complete all test items, results from 236 individuals were included in the final analyses ($M_{age}$ = 49.51, $SD_{age}$ = 15.49).

Factor structure of the proposed model was analyzed using exploratory factor analysis (EFA) with Oblimin rotation of principal components. Test score reliability (Cronbach's alpha) and model fit indices were examined: Kaiser-Meyer-Olkin test (KMO), chi-square ($\chi^2$), root mean square error of approximation (RMSEA), comparative fit index (CFI). KMO values higher than 0.80 show an acceptable sampling adequacy. RMSEA values below 0.08 indicate acceptable fit, and values above 0.10 indicate poor fit; for CFI values higher than 0.90 show an acceptable model fit.

As a result of exploratory factor analysis single factor, prayer importance, explaining 66.05% of variance of the construct, was identified (KMO = 0.897, $\chi^2$ = 755.15, df = 15, $p < 0.001$). Items of the scale showed high levels of internal consistency which indicated

that they measure the same phenomenon ($\alpha = 0.896$). Exploratory structural equation modelling (ESEM) was conducted to verify the construct validity of the proposed model (Figure 1).

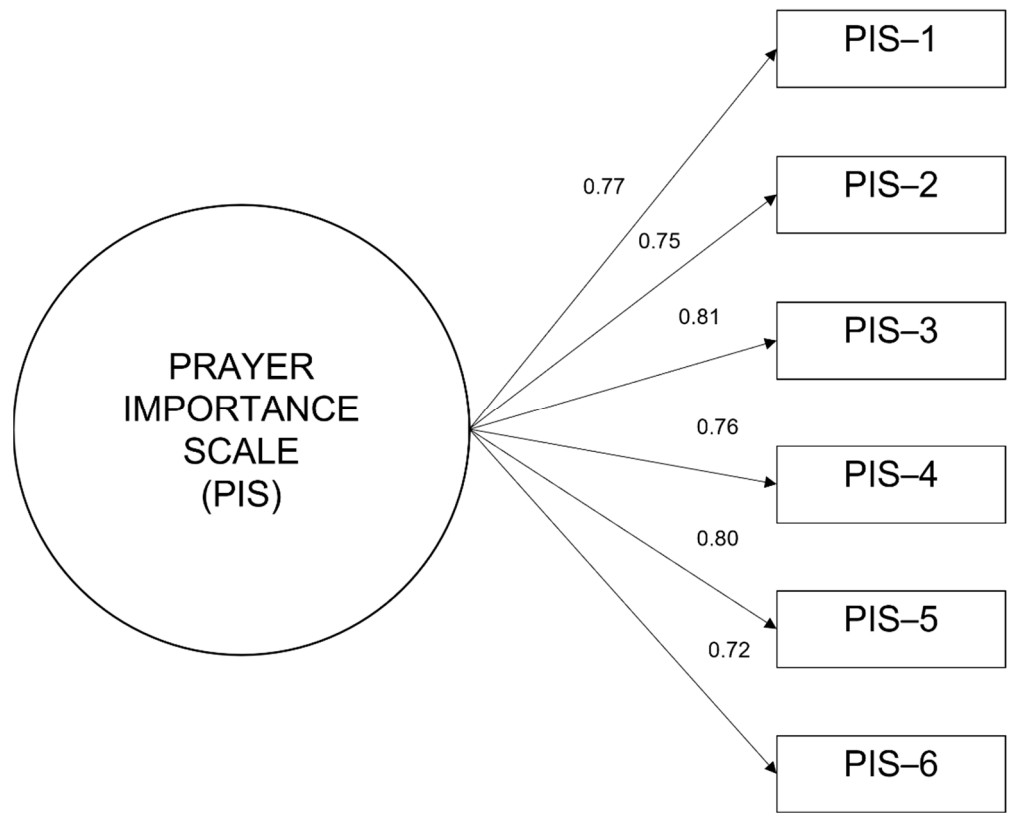

**Figure 1.** ESEM of PIS (regression weights)—study 1. Source: own elaboration.

All test items were found to correlate at a very high level with the latent variable, prayer importance. Model fit indices were observed to partially explain model fit. CFI indicated an acceptable fit, the RMSEA index was on the line of acceptability (see Table 2).

**Table 2.** Fit indices in Study 1.

| Chi-Square | df | $p$ | Pcmin/df | CFI | RMSEA | LO90 | HI 90 | PCLOSE |
|:---:|:---:|:---:|:---:|:---:|:---:|:---:|:---:|:---:|
| 260.43 | 9 | 0.002 | 20.94 | 0.977 | 0.091 | 0.052 | 0.132 | 0.044 |

To assess the concurrent validity of the method it was correlated with Structure and Level of Religiosity Test (SLRT) (Rydz et al. 2017) and an external criterion—frequency of the Holy Communion (see Table 3).

**Table 3.** Correlation between PIS and SLRT—study 1.

| | PIS | Awareness | Feelings | Decisions | Bond | Practices | Morality | Experiences |
|:---|:---:|:---:|:---:|:---:|:---:|:---:|:---:|:---:|
| Awareness | 0.445 ** | | | | | | | |
| Feelings | 0.519 ** | 0.681 ** | | | | | | |
| Decisions | 0.317 ** | 0.412 ** | 0.458 ** | | | | | |
| Bond | 0.391 ** | 0.423 ** | 0.407 ** | 0.361 ** | | | | |
| Practices | 0.283 ** | 0.410 ** | 0.398 ** | 0.373 ** | 0.387 ** | | | |
| Morality | 0.384 ** | 0.595 ** | 0.512 ** | 0.392 ** | 0.428 ** | 0.649 ** | | |
| Experiences | 0.458 ** | 0.571 ** | 0.513 ** | 0.362 ** | 0.370 ** | 0.556 ** | 0.690 ** | |
| Forms | 0.258 ** | 0.389 ** | 0.309 ** | 0.378 ** | 0.327 ** | 0.313 ** | 0.372 ** | 0.354 ** |

Legend: ** $p < 0.01$.

The correlation between PIS and an external criterion—frequency of the Holy Communion turned out to be positive, statistically significant and moderate (Spearman's $\rho$ = 0.35, $p < 0.001$).

### 2.2. Study 2

In the second study, it was verified whether the proposed model showed satisfactory fit indices in a new sample of participants. Because three individuals did not complete all test items, results from 217 individuals were included in the final analyses ($n_{women}$ = 107) ($M_{age}$ = 48.34, $SD_{age}$ = 16.48). Confirmatory factor analysis (CFA) was conducted to verify construct validity of the proposed model (Figure 2).

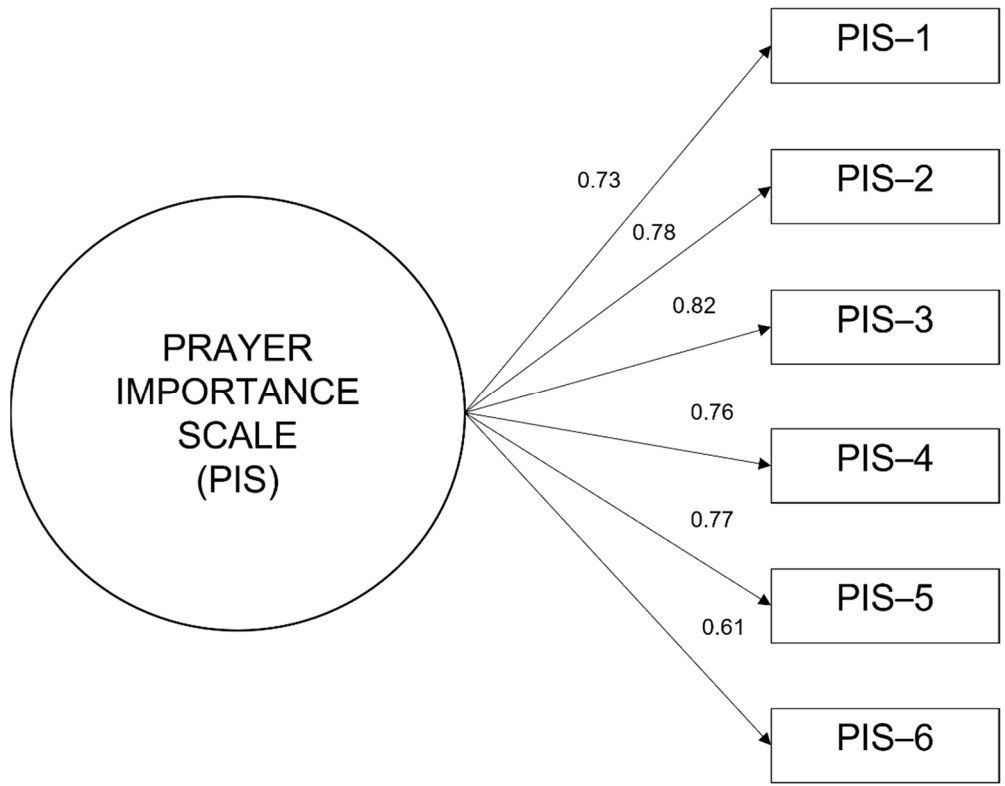

**Figure 2.** CFA of PIS (regression weights)—study 2. Source: own elaboration.

Similar to study 1, high correlations of the test items with the latent variable, prayer importance were observed. Fit indices confirmed good model fit to the data (see Table 4).

**Table 4.** Fit indices of the model in study 2.

| Chi-Square | df | $p$ | Pcmin/df | CFI | RMSEA | LO90 | HI 90 | PCLOSE |
|---|---|---|---|---|---|---|---|---|
| 60.807 | 9 | 0.657 | 0.756 | 1 | 0.000 | 0.000 | 0.062 | 0.897 |

To determine validity of the scale it was correlated with SLRT (Rydz et al. 2017) and an external criterion–frequency of the Holy Communion (see Table 5). The correlation turned out to be positive, statistically significant and moderate (Spearman's $\rho$ = 0.37, $p < 0.001$). Therefore, the findings of study 1 were confirmed.

**Table 5.** Correlations between PIS and SLRT—study 2.

|  | PIS | Awareness | Feelings | Decisions | Bond | Practices | Morality | Experience |
|---|---|---|---|---|---|---|---|---|
| Awareness | 0.283 ** | | | | | | | |
| Feelings | 0.339 ** | 0.562 ** | | | | | | |
| Decisions | 0.266 ** | 0.322 ** | 0.444 ** | | | | | |
| Bond | 0.352 ** | 0.435 ** | 0.330 ** | 0.465 ** | | | | |
| Practices | 0.405 ** | 0.503 ** | 0.468 ** | 0.371 ** | 0.438 ** | | | |
| Morality | 0.318 ** | 0.520 ** | 0.431 ** | 0.436 ** | 0.480 ** | 0.636 ** | | |
| Experience | 0.368 ** | 0.547 ** | 0.501 ** | 0.463 ** | 0.454 ** | 0.660 ** | 0.638 ** | |
| Forms | 0.344 ** | 0.324 ** | 0.448 ** | 0.506 ** | 0.576 ** | 0.431 ** | 0.460 ** | 0.373 ** |

Legend: ** $p < 0.01$.

## 3. Discussion

Prayer is an expression of spiritual activity for many religions–primarily for monotheistic religions, such as Christianity, Judaism and Islam, but it also has equivalents in Buddhism and Hinduism (e.g., meditation). Each person's concept of prayer seems to correspond to a large extent to his or her past, upbringing, personality, as well as education, cognitive, emotional and social functioning (Walesa and Tatala 2020). Current research on prayer is conducted along two paths. On one hand it is focused on personality traits and individual differences. On the other, there are attempts to create coherent definitions and operationalizations of prayer to get to know the theoretical foundations of prayer and at the same time to derive a common or methodologically reconcilable apparatus which will make it possible to compare conclusions coming from different studies (Ladd and McIntosh 2008).

The aim of this study was to validate the psychometric properties of the PIS, which was developed for a rapid diagnosis of religious people. Many definitions as well as types of prayer (e.g., petition, worship, and thanksgiving) were distinguished. So far, however, there has been no method which would implicitly control importance of the role prayer takes in a person's life Besides the time devoted to prayer, it is the subjective importance of prayer that can constitute an additional criterion of its role in human life. In this study it was operationalized by several criteria that relate to persisting in prayer regardless of external factors, such as setbacks and obligations (Tatala 2009; Cornwall et al. 1986).

A 6-item scale was extracted based on the premise that prayer is central to religiosity. Two studies were conducted: the first included exploratory factor analysis (CFA), reliability testing and correlating the presented measure with the Structure and Level of Religiosity Test (SLRT) and frequency of the Holy Communion. Satisfactory properties of fit indices measures (CFI, RMSEA, PCLOSE) were obtained in study 1 which confirmed the proposed model. In the second study, the conclusions of study 1 were upheld: correlations of PIS with basic parameters of religiosity: religious awareness, religious feelings, religious decisions, bond with a fellowship of believers, religious practices, religious morality, religious experience and forms of profession of faith were found to be significant. Walesa's conception proposes that prayer as a component of forms of profession of faith constitutes the core of religiosity. Thus, correlation of PIS and aforementioned parameters may account for the high usefulness of the method for the initial diagnosis of religiosity. Moreover, PIS was observed to correlate positively with an objective indicator of religiosity (in the case of Christian religion), frequency of the Holy Communion. This indicator was perceived as relevant for the sample of Catholics in Poland–due to the criterion of ecological accuracy.

Until now, the subject of prayer importance has played a minor role. Huber (Huber and Huber 2012) in the Centrality of Religiosity Scale (CRS), expressed its subjective importance is in terms of a single item (How important is personal prayer for you?), referring to the importance of prayer explicitly, and not verifying its manifestations. The topic was indirectly raised by Walesa and colleagues in SLRT, but test items (When God does not hear my prayers, I feel rebellion against Him, I do not pray for the dead) address a narrow theoretical scope of the subject.

One advantage of the presented method is that it is of universal character and can be used with people of different faiths. The method can also be used successfully in pastoral counselling, which undoubtedly constitutes its application value. In future, it would be interesting to see if individuals scoring high on the PIS are also characterized by high levels of adaptive situational coping strategies. In addition, investigating mechanisms underlying the importance given to prayer would be worthwhile as prayer may be an important moderator in relationships e.g., between prayer and quality of life, life satisfaction, or healing.

**Supplementary Materials:** The following are available online at https://www.mdpi.com/article/10.3390/rel12111032/s1.

**Author Contributions:** Conceptualization, M.T.; Formal analysis, M.W.; Investigation, M.T.; Methodology, M.T. and M.W.; Project administration, M.T.; Supervision, M.T.; Visualization, M.W.; Writing—original draft, M.T.; Writing—review & editing, M.T.; Funding acquisition, M.T. All authors have read and agreed to the published version of the manuscript.

**Funding:** This research received no external funding. The APC was funded by The John Paul II (The Second) Catholic University of Lublin.

**Institutional Review Board Statement:** Ethical review and approval were waived for this study, due to its minimal risk to participants.

**Informed Consent Statement:** Informed consent was obtained from all subjects involved in the study.

**Data Availability Statement:** The data presented in this study are available in supplementary materials. The data in extended version are available on request from the corresponding author.

**Conflicts of Interest:** The authors declare no conflict of interest.

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
