# Peer review of "The Validity of Prayer Importance Scale (PIS)"

_religions, doi:10.3390/rel12111032_

Round 1

Reviewer 1 Report

I don't have a lot to say about this excellent article. There were a couple of abrupt transitions in section one. 

The methodology and analysis are sound. They followed logical steps to develop the scale, and they conducted two different experiments to validate the scale. This scale contributes to the study of prayer, and it should be widely used in the future. 

Author Response

We are very grateful for this remark. In fact, our aim was to conduct two experiments to create a method that would be a useful contribution to the study of prayer. Therefore, we hope that method (PIS) will be successfully applied in the future.

Reviewer 2 Report

line 77 Repine?

Numbering of items at ll. 101-108 is unclear. 1. is the title of the list of six test items!

If the list reorganizes the items in decreasing indices value, why is 'Although other events...' in second position while 'Even though...' has a higher M value1?

Holy Communion is not exactly an objective indication of religiosity in Christian religion (line 197). May be correct for Polish Catholicism, less so in other types of Catholicisms and Orthodoxy and quite misleading for Protestants of many shades.

What practical use do you envisage for the ability to measure religiosity with PIS?

Author Response

We greatly appreciate your comments. We addressed them one by one:

  1. line 77 Repine?
    1. One of the components mentioned in the Huber model is repine, it means complaining, expressing your dissatisfaction (feeling or expressing something like upset, being unhappy and miserable) (line 77)
  2. Numbering of items at ll. 101-108 is unclear. 1. is the title of the list of six test items!
    1. Thank you for your comment. We made the appropriate corrections in the text (lines 101-108). It is likely that the mistake occurred while formatting the document.
  3. If the list reorganizes the items in decreasing indices value, why is 'Although other events...' in second position while 'Even though...' has a higher M value?
    1. Thank you for this comment. The list did not reorganise the test items by decreasing mean. The test items are written out sequentially according to the order in which they appeared in the questionnaire. Sometimes the order of the items is important, so we have left them as they were presented to the assessors. Otherwise it would seem that they rated the first items highest and all subsequent items lower and lower.
  4. Holy Communion is not exactly an objective indication of religiosity in Christian religion (line 197). May be correct for Polish Catholicism, less so in other types of Catholicisms and Orthodoxy and quite misleading for Protestants of many shades.
    1. Thank you for this comment. We have added the appropriate notation in line 197. We have indicated that Holy Communion is one (but not the only one) of the indicators that can be used (at least in Poland, to keep the ecological criteria for the research sample)
  5. What practical use do you envisage for the ability to measure religiosity with PIS?
    1. We have included (line 209-211) information about practical application in the text. Thank you for this comment.

Reviewer 3 Report

Comments:

  1. The research aims to provide a universally accepted instrument to determine participants' or subjects' religiosity.
  2. The instrument fills research gaps, especially the insufficient scholarship on hard-to-measure latent behaviors associated with religiosity or what the author posits as the subjective components/elements of prayer.
  3. The author emphasizes the power and importance of prayer in people's lives.
  4. The manuscript can be made richer by incorporating specific examples or case studies.

Specific comments:

  1. Lines 189-193 can be transferred to the top of the manuscript to clarify and make explicit the study's goal.
  2. I am not entirely convinced why there is a need to create an independent psychometric tool on prayer. The other research mentioned by the author seemed to have already captured the essence of prayer.  Moreover, prayer does not only consist of sets of beliefs that are associated with spirituality or religion. Prayer is also a performance.  How does a psychometric tool capture the bodily dimensions of prayer?
  3. Lines 168-169 need to be further explained by citing instances or past studies relating prayer to a person's past or cognitive capacity. In the earlier part of the manuscript, the author referred to studies that manifested that the prayerful are often the economically challenged in society.
  4. Line 178, "So far, however, there has been no method which would present the subjective prayer importance." Can the author further explain what these subjective dimensions/elements of prayer are that they refer to in this claim? Moreover, why is it essential to capture the subjective dimensions of prayer?
  5. Regarding the author's recommendation that the PIS can be helpful in Hinduism, Buddhism, and Islam, can the author cite specific examples of the subjective elements in the prayers of the adherents of the religions mentioned earlier?

Author Response

We are very grateful for your comments. We have addressed them one by one:

  1. Lines 189-193 can be transferred to the top of the manuscript to clarify and make explicit the study's goal.
    1. Thank you for this comment. We have moved the conclusions about the correlation of the PIS scale with Walesa's religiosity scales to the abstract (lines 8-11).
  2. I am not entirely convinced why there is a need to create an independent psychometric tool on prayer. The other research mentioned by the author seemed to have already captured the essence of prayer.  Moreover, prayer does not only consist of sets of beliefs that are associated with spirituality or religion. Prayer is also a performance.  How does a psychometric tool capture the bodily dimensions of prayer?
    1. Thank you for this comment. We believe that there is a need for an independent tool on prayer. One of the reasons is its application in pastoral counselling - we included information about this in the text (line 209-211). We agree that prayer, as a multidimensional phenomenon, is to some extent elusive, and to ignore it and not attempt to study it would result in an over-reductionist view of different kinds of motivation that guide human life. Of course, we are aware our method is in some sense at a tip of an iceberg, even though its core and strength is operationalization of prayer as something that can play a great role in human life. The nature of prayer itself was not the direct focus of our research, but we wanted to show that with our method we can verify whether prayer actually plays an important role in someone's life. Commonly, we ask a person how often they pray and then move on to the types of prayer. We pointed out that it is a rarity to ask whether prayer is in fact an important activity for this person.
  3. Lines 168-169 need to be further explained by citing instances or past studies relating prayer to a person's past or cognitive capacity. In the earlier part of the manuscript, the author referred to studies that manifested that the prayerful are often the economically challenged in society.
    1. Thank you for this comment. We have added citations to support the mentioned relationships (lines 168-169).
  4. Line 178, "So far, however, there has been no method which would present the subjective prayer importance." Can the author further explain what these subjective dimensions/elements of prayer are that they refer to in this claim? Moreover, why is it essential to capture the subjective dimensions of prayer?
    1. We agree with this remark. In fact, the wording could be perceived by the reader as quite unclear. By prayer importance we meant importance in their life. The sentence sounds too laconic and general, so we decided to change the mentioned phrase to "So far, however, there has been no method which would implicitly control importance of the role prayer takes in a person's life.” (lines 179-181)
  5. Regarding the author's recommendation that the PIS can be helpful in Hinduism, Buddhism, and Islam, can the author cite specific examples of the subjective elements in the prayers of the adherents of the religions mentioned earlier?
    1. By the sentence stating that PIS can also be useful in investigating people who follow the above religions, we rather meant that no matter the type of religion, prayer can always have some role in a person's life. In other words, a person can have some kind of relation toward prayer. It was not in our interest to enter into the elements of any religion understood in a direct way. We chose Polish Catholics as our target sample, but only because they are in the majority in Poland and there was no problem with reaching this sample. We do not conceal that it would indeed be worthwhile in the future to use the PIS method and check the factor structure also in case of other religions. However, during the construction of the method we were guided by the aim that it could be used regardless of the type of religion - so as to examine only the importance it plays in people's lives.

Reviewer 4 Report

Very good work.

Author Response

Thank you for your positive assessment of our paper.